# Human Cytomegalovirus Immune Evasion of Natural Killer Cells: A Virus for All Seasons?

**DOI:** 10.3390/pathogens14070629

**Published:** 2025-06-24

**Authors:** Hannah Preston, Rowan Casey, Elizabeth Ferris, Lauren Kerr-Jones, Lauren Jones, Farah Latif, Mathew Clement, Rebecca J. Aicheler, Eddie C. Y. Wang, Richard J. Stanton, Ceri A. Fielding

**Affiliations:** 1Division of Infection and Immunity, School of Medicine, Cardiff University, Cardiff CF14 4XN, UK; prestonh2@cardiff.ac.uk (H.P.); caseyr1@cardiff.ac.uk (R.C.); ferrisem1@cardiff.ac.uk (E.F.); kerrle@cardiff.ac.uk (L.K.-J.); jonesl226@cardiff.ac.uk (L.J.); latiffe@cardiff.ac.uk (F.L.); clementm@cardiff.ac.uk (M.C.); 2Systems Immunity University Research Institute (SIURI), Cardiff University, Cardiff CF14 4XN, UK; 3School of Sport and Health Sciences, Cardiff Metropolitan University, Cardiff CF5 2YB, UK; raicheler@cardiffmet.ac.uk

**Keywords:** cytomegalovirus, natural killer cell, immune evasion

## Abstract

Human cytomegalovirus (HCMV) is a ubiquitous member of the herpesvirus family, of significant clinical importance, and highly adapted to its host, resulting from millions of years of co-evolution. As a result, the virus systematically subverts almost all aspects of antiviral immune defence to successfully establish a lifelong persistent infection, and in the process, dramatically reshapes the phenotype and function of host immunity to both HCMV and other diseases. Natural killer (NK) cells are a critical component of successful herpesvirus control. Here, we discuss their role in modulating HCMV disease and the multitude of ways that HCMV has evolved to prevent and manipulate this process. We also consider how antibody-dependent cellular cytotoxicity by NK cells directed against HCMV might overcome NK immune evasion mechanisms and be useful therapeutically.

## 1. Human Cytomegalovirus (HCMV)

Human cytomegalovirus (HCMV) is a betaherpesvirus with global health implications that is well known for its extensive immune evasive properties. Although not amongst the most contagious viruses [1], HCMV is ubiquitous in human populations worldwide. In the developing world, the majority of children are HCMV seropositive by the age of 18 months [2]. While in the developed world (e.g., USA), acquisition occurs throughout life, with seropositivity rising from ~31% at 4 years of age and under to approximately 70% by adulthood [3,4]. Natural transmission occurs through bodily fluids such as saliva, breast milk, and urine [5] and also sexually [6]. Like all herpesviruses, HCMV establishes a lifelong latent infection. For HCMV, the latent reservoir is thought to be CD34^+^ haemopoietic progenitor stem cells, with virus reactivation occurring following later differentiation of monocytes into macrophages and dendritic cells [7]. Lytic infection is also supported in a broad range of non-hematopoietic cell types, including epithelial cells, endothelial cells, and fibroblasts. Unusually, immunity raised to primary infection is insufficient to prevent reinfection. Thus, individuals can harbour multiple HCMV strains [8,9,10], and disease can be caused by primary infection, reactivation of the primary isolate, or reinfection with a second isolate [11]. As a result, HCMV evolution is characterised by extensive recombination [12].

For immunocompetent individuals, robust antiviral immune responses normally prevent the development of symptoms during primary infection [13]. However, in cases of immunological naïvety and/or suppression, weaker immune responses result in a failure to control infection and severe disease can occur. Congenital cytomegalovirus (cCMV) infections occur in ~0.67% of births [14]. Fetal death can occur, and approximately 13% of live cCMV births will present with sensorineural hearing loss (SNHL) and neurodisabilities. A significant number of cases will develop symptoms in the months and years after birth, with a final burden of up to 20% of cases being symptomatic [15]. Importantly, cCMV can be caused by both reactivation and reinfection. The burden in developing countries (where seroprevalence at pregnancy approaches 100%) is equivalent to that in the developed world (where seroprevalence at pregnancy is significantly lower). This equivalence perhaps indicates that although naturally acquired pre-existing immunity may lower incidence, it is insufficient to completely prevent detrimental cCMV outcomes [15,16].

In immunocompromised patients such as transplant recipients, some viraemic patients develop end-organ disease (EOD), graft-versus-host disease (GVHD) and opportunistic infections, with pathologies varying depending on the affected organs. If untreated, these infections lead to high mortality [17], so patients are therefore treated with antivirals such as valganciclovir and letermovir. However, treatment can lead to the development of antiviral-resistant strains. Immunosuppressive drug regimens can also be reduced to restore some immunological control; however, this risks allograft rejection in transplant patients [13,18]. In the solid organ transplant setting, risk is highest when a seronegative individual is in receipt of an organ from a seropositive donor. In the stem cell setting (where immunity is reconstituted from the donor), the risk is highest when a seropositive recipient is transplanted with cells from a seronegative donor. Thus, in both cases, pre-existing immunity reduces disease risk but is not preventative.

HCMV proteins and DNA have also been detected in various tumours, including glioma, medulloblastoma, neuroblastoma, colorectal cancer, prostate cancer and breast cancer [19,20,21]. It is currently unclear whether HCMV has a causal role in oncogenesis or whether it infects existing tumours and influences disease progression through oncomodulation. Longitudinal studies in a large cohort are necessary to define any causal link akin to those that have linked Epstein-Barr virus infection to the development of multiple sclerosis [22].

Herpesviruses have co-evolved with their hosts for at least 180–220 million years [23,24], with parallel divergence occurring in the viruses as hosts have undergone speciation. As a result, herpesviruses are normally highly species-specific, with many species carrying their own herpesvirus repertoire. HCMV has the largest genome of any human virus at ~236 kilobase pairs [25]. It encodes approximately 170 canonical protein-coding open-reading frames (ORFs), of which only 45 are essential for replication in fibroblasts *in vitro* [25,26]. The remaining 125, while non-essential *in vitro*, are presumably important for virus transmission and persistence in the human host. The combination of a high coding capacity, the ability to establish lifelong infection, and long co-evolution with its host has permitted HCMV to develop an exceptionally broad range of mechanisms to ensure survival despite inducing one of the largest and most multifaceted immune responses to any pathogen [27]. Thus, many of the ‘non-essential’ genes have important roles in evading different aspects of the immune system, including intrinsic immunity, innate immunity, T-cells, and natural killer (NK) cells [28,29,30,31]. We focus here specifically on the NK response.

## 2. Natural Killer Cell Activation—Innate Immunity

In humans, natural killer (NK) cells are classified as CD3^−^ CD56^+^ lymphoid cells. They are involved in both innate and adaptive immunity by secreting proinflammatory cytokines and releasing cytotoxic granules to lyse virally infected or cancerous cells. During the innate immune response, NK cells can be activated by Type 1 interferons, IL-2, IL-12, IL-15, IL-18, IL-21 and IL-27 [32,33,34,35,36,37,38], which are secreted by infected or antigen-presenting cells. Alternatively, NK cells can be activated through the balance of signals from their varied activating or inhibitory natural killer cell receptors (NKRs). These are subdivided into various families, some of which include ‘paired’ immune receptors, in which an activating and inhibitory receptor(s) recognise a common ligand. Killer immunoglobulin-like receptors (KIRs) are a family of highly polymorphic activating and inhibitory NKRs that surveil human leukocyte antigen (HLA-I) [39]. Viruses such as HCMV downregulate endogenous HLA-I to prevent CD8^+^ T-cell activation. This loss is detected through a reduction in inhibitory KIR signalling that would normally lead to robust NK activation and control of infection [40]. Natural cytotoxicity receptors (NCRs) are NKRs that can bind to a range of stress, infection, or transformation-induced ligands on infected cells, leading to activation. For example, NKp30 binds to B7-H6, while NKp44 recognises mixed-lineage leukaemia protein 5 (MLL5)/NKp44L, HLA-DP, Nidogen-1, Globo-A and proliferating cell nuclear antigen (PCNA) [41,42,43,44,45,46,47]. Stochastic expression of this wide array of receptors means that there may be up to 30,000 different NK phenotypes in any one donor [48]. Despite this variation, the bulk NK population in peripheral blood are referred to as conventional NK cells (cNK), whereas specific NK phenotypes are referred to by other names, e.g., adaptive or tissue-resident NK cells [49]. Irrespective of drivers of NK activation, sustained stimulation leads to robust upregulation of oxidative phosphorylation, glycolysis and antiviral functionality [50].

Classically in the blood, NK cells can be divided into 3 subsets based on their CD56 and CD16 expression levels: CD56^bright^ CD56^dim/−^, CD56^dim^ CD16^bright^ and CD56^dim^ CD16^−^. In the peripheral blood, ~90% of NK cells are CD56^dim^ CD16^bright^, while CD56^bright^ CD16^dim/−^ NK cells account for much of the remainder [51], although phenotypes of tissue-resident NK cells may differ [52]. CD56^bright^ CD16^dim^ NK cells predominately secrete proinflammatory cytokines, including IFN-γ and TNFα, to enhance and coordinate innate and adaptive immune responses [53]. In contrast, CD56^dim^ CD16^bright^ NK cells are predominately cytotoxic. They degranulate to release perforin, which creates pores in the cell membrane of a target infected cell, and granzymes, which enter the cell and cleave caspases, thereby inducing apoptosis [54].

## 3. Natural Killer Cell Activation—Adaptive Immunity

In addition to being the major cellular component of innate immunity, NK cells also play important roles as a component of adaptive immunity. Most circulating NK cells express CD16 [51,55]. CD16 is an activating Fc gamma receptor (FcγR), also known as FcγRIIIa, and is the main activating FcγR expressed in NK cells [56]. CD16 binds with low affinity to the Fc portion of IgG following the coating of a cell, resulting in activation of the NK cell and lysis of the target cell [57,58]. This is called antibody-dependent cellular cytotoxicity (ADCC). CD16 is unique among NK receptors in being able to trigger resting NK cells in isolation without the involvement of any other receptors [59]. Multiple studies have demonstrated that ADCC is a critical component of immunological defence against both viruses and tumours [60]. However, HCMV is unique in driving the expansion of a subset of NK cells that are exceptionally effective at carrying out ADCC [61,62,63]. These cells share qualities with cytotoxic T cells and are often termed “adaptive” NKs [64]. Although they expand in response to HCMV infection, their altered properties are not restricted to HCMV, i.e., responses are enhanced to any antibody-coated target, including both virus-infected and transformed cells [65,66]. They are seen to varying degrees in most healthy adults with latent HCMV infection, as well as in organ-transplant recipients who subsequently experience HCMV infection and reactivation [67]. In these cases, the adaptive NK cell population persists past the point of acute infection and continues to proliferate for at least 1-year post-transplantation [68].

Phenotypically, adaptive NK cells (as opposed to their conventional counterparts, cNKs) are commonly CD16^+^, NKG2C^+^, NKG2A^−^, CD57^+^ and CD56^bright^ [69]. Several additional markers are used to define these cells, including a lack of cell surface NKp30 and intracellular FcεRγ, and restricted KIR expression [70]. Furthermore, these NK cells can promote the recruitment of T-cells and release an extensive range of cytokines and chemokines, all of which may aid virus control [71]. It is likely that adaptive NK cells have enhanced ADCC capability due to an increase in the expression of co-receptors and improved receptor signalling. For example, whereas cNKs express two CD16 adapter molecules, namely CD3ζ, a component of the T-cell receptor complex, and FcεRγ, adaptive NK cells only express CD3ζ. CD3ζ and FcεRγ differ in their number of immunoreceptor tyrosine-based activation motifs (ITAMs) (CD3ζ has 3 ITAMs, whereas FcεRγ has one), as well as their capacity to support the expression of other receptors such as NKp30, potentially explaining some of the functional differences observed [72]. The change in adapter and receptor expression patterns in adaptive NK cells is underpinned by epigenetic modifications of genes associated with antibody-dependent responses, potentially to adapt to the chronic challenge of HCMV, enhancing ADCC efficiency over time [73,74,75,76]. These cells can also be expanded from peripheral blood lymphocytes *in vitro* when stimulated with HCMV-infected fibroblasts and can eventually outnumber NK cells expressing CD94/NKG2A, the inhibitory “paired” counterpart of CD94/NKG2C [77]. Upregulation of cell surface HLA-E by the signal peptide of HCMV UL40 may also be involved in their expansions (see Section 6.2 for more details) [78].

## 4. Importance of NK Cells in Controlling HCMV Infection

NK cells are critical for the control of HCMV, as evidenced by NK defects, either due to primary immunodeficiencies or acquired conditions. These significantly influence the body’s ability to manage CMV infection, leading to increased morbidity and/or mortality. Genetic syndromes such as *GATA2* deficiency or mutations in genes that lead to loss of CD16 (*MCM4* and *FCGR3A*) have been linked to impaired NK cell development and function and are commonly associated with severe or atypical HCMV infections [79,80]. These cases highlight the essential role of NK cells in early viral recognition and cytotoxic activity through both perforin/granzyme release and ADCC.

In the context of hematopoietic stem cell transplantation (HSCT), the presence and early reconstitution of functional NK cells are associated with improved CMV control and enhanced survival outcomes [81]. In addition, recent approaches that enhance NK cell function, including cytokine-based NK cell activation, have been shown to induce remission in lymphodepleted acute myeloid leukaemia patients without ensuing graft versus host disease [82,83,84], underscoring the *in vivo* anti-HCMV efficacy of NK cells.

Multiple studies also point specifically to the role of adaptive NK cells in the control of infection and/or disease. Patients with larger adaptive NK expansions tend to express higher amounts of IL-15, IFN-γ and T-bet and show better virus control, with NK cells capable of controlling infection even in the absence of T-cell responses [68,85]. In HSCT, NKG2C^+^ NK cells are expanded following CMV reactivation in all cases, but patients receiving transplants from seropositive donors exhibited heightened antiviral responses to HCMV compared with those from seronegative donors. This suggests that adaptive NK cells may be transferred to the transplanted organ and are better at controlling viruses [81]. Similarly, kidney transplant recipients had remarkably reduced incidences of posttransplant viremia when pretransplant levels of NKG2C^+^ NK cells were higher, where reduced frequencies of these NK cells were associated with the onset of symptomatic HCMV infection [86]. Studies focusing on HCMV reactivation following the transplantation of both solid organs or stem cells found that higher numbers of NKG2C^+^ NK cells in the recipient were inversely correlated with HCMV reactivation and HCMV blood titres. This suggests that the number of circulating adaptive NK cells directly influenced patient prognosis and clinical outcomes post-transplantation [87,88,89,90,91]. In the context of cCMV, foetuses that acquire HCMV *in utero* display long-term expansions of NKG2C^+^ NK cells [92,93,94]. Furthermore, pre-existing maternal immunity resulting from non-primary infection is associated with protection against intrauterine transmission of CMV infection [95,96]. The circulating NK cells from non-transmitting mothers showed typical characteristics of an adaptive NK cell phenotype compared with transmitting mothers who had more cNK cells [97]. Together, these studies highlight that adaptive NK cells (and, by extension, ADCC) may be a key component of successful HCMV control *in vivo*.

Finally, the fact that HCMV dedicates such a large proportion of its genome to encoding many diverse immune evasion mechanisms that specifically target both innate and adaptive NK cell functions (see below) emphasises the ability of fully functional NK cell responses to provide potent viral control, which must be effectively counteracted by the virus.

## 5. Broad Mechanisms of NK Immune Evasins

Broadly speaking, immune evasion mechanism strategies employed by HCMV to subvert NK cell activation can be grouped into eight categories (Figure 1). These are (i) expression of virus-encoded cell surface ligands for inhibitory NK receptors, (ii) upregulation of endogenous cell surface ligands for inhibitory NK receptors, (iii) downregulation of host cell surface ligands for activating NK receptors by intracellular retention, or (iv) proteasomal or lysosomal degradation, (v) impairment of immune synapse formation by targeting the actin cytoskeleton or adhesion molecules, (vi) impairment of death receptor interactions, (vii) inhibition/skewing of NK cell differentiation, and (viii) encoding viral Fc receptors to antagonise ADCC.

HCMV NK immune evasion can be mediated by either the upregulation of endogenous or virus-encoded ligands for inhibitory NK receptors or the downregulation of endogenous ligands for activating receptors. The latter may take place through either intracellular retention or degradation via the proteasome or lysosome. HCMV also impairs NK cell-killing mechanisms by inhibiting cell death pathways at the ligand-receptor level or at the level of downstream caspase activation. It also interferes with the formation of the immunological synapse by targeting adhesion molecules or actin polymerisation.

## 6. Specific HCMV NK Immune Evasin Mechanisms

### 6.1. HCMV-Encoded HLA-I Homologues

LIR-1 (also known as LILRB1, ILT2, or CD85j) is an inhibitory NK receptor that normally receives an inhibitory signal from HLA-I. Since HLA-I is downregulated on HCMV-infected cells, HCMV compensates by expressing gpUL18, a HLA-I homolog that binds to and elicits an inhibitory signal in NK cells through LIR-1 (Figure 2) [98,99]. Like HLA-I, gpUL18 is expressed as a trimeric complex with beta-2 microglobulin and peptide [100,101]. gpUL18 binds LIR-1 with a 1000-fold greater affinity than HLA-I, thereby out-competing HLA-I [98,102]. gpUL18 shares ~25% sequence identity with HLA-I. In contrast to HLA-I, which has only one N-glycan site, gpUL18 is heavily glycosylated with 13 potential N-glycosylation sites [103,104]. However, elegant structural studies showed that the gpUL18-LIR-1 complex shares similarities with the HLA-A2-LIR-1 and HLA-G-LIR-1 complexes [104,105]. Cell surface expression of gpUL18 is enhanced by the signal peptide of UL40, which also up-regulates HLA-E (Section 6.2) [99]. Furthermore, whilst the UL18 sequence is not highly variable, genetic variation does occur in clinical HCMV strains, and these variants exhibit altered abilities to bind to LIR-1 [105,106,107,108]. Additionally, genetic variation in LIR-1 alters the affinity of LIR-1 for gpUL18 but not for HLA-I, and transplant patients with a lower frequency of LIR-1^+^ NK cells are disadvantaged in controlling HCMV upon immunosuppression [109,110]. Originally characterised as an inhibitor of LIR-1^+^ NK cells, evidence also indicates that gpUL18 is able to stimulate LIR-1^−^ NK cells, suggesting a capacity to modulate the NK cell response via a yet-to-be-defined mechanism [99,110].

HCMV encodes an HLA-I mimic, gpUL18, which binds to an inhibitory receptor, LIR-1, on NK cells. The signal peptide of UL40 (spUL40) has homology with the leader sequence of HLA-I and is able to upregulate cell surface expression of HLA-E in a TAP-independent manner. HLA-E can bind the inhibitory CD94/NKG2A complex or its activating counterpart, CD94/NKG2C. spUL40 also up-regulates cell surface expression of gpUL18.

### 6.2. Upregulation of Endogenous HLA-E Cell Surface Expression

The non-classical HLA molecule HLA-E is upregulated at the cell surface following binding to conserved peptides derived from the leader sequences of other HLA-I molecules. Once there, it interacts with paired CD94/NKG2A and/or CD94/NKG2C receptors, thereby providing surveillance for endogenous HLA-I levels. Due to virus-induced targeting of the transporter associated with antigen processing (TAP) and subsequent loss of the native HLA-derived peptide, HCMV encodes a protein with a peptide mimic in its signal peptide (spUL40), which upregulates HLA-E independently of TAP (Figure 2) [111,112,113]. HLA-E bound spUL40 peptides modulate NK cell responses through direct binding to either CD94^+^NKG2A^+^ or CD94^+^NKG2C^+^ [78]. In HCMV seronegative individuals, CD94^+^NKG2C^+^ cells comprise 1–4% of peripheral blood NK cells, while in HCMV seropositive individuals, CD94 + NKG2C + cells may comprise up to 22.1% of the NK cell repertoire [70]. Importantly, the CD94^+^NKG2C^+^NK cells in HCMV seropositive individuals are phenotypically and functionally distinct from those in HCMV seronegative individuals (See Section 3 and Section 7) [70,74,75,114], and the expansion of adaptive NK cells in HCMV seropositive individuals can be influenced by NKG2C interactions with the HLA-E/UL40 peptide complex [78]. In addition, allelic variation exists in both NKG2C [115] and HLA-E, with those in HLA-E capable of subtly modulating the avidity of recognition by CD94/NKG2 receptors [116]. Together, genetic variation in the viral UL40 leader sequence, NKG2C, and HLA-E all have the capacity to impact the frequency and function of the circulating repertoire of NK cells in seropositive individuals [78,117]. The existence of UL40 variations implies a complex interplay between host and virus in which the virus maintains diversity in an immune evasin to counteract host diversity. However, the range of situations in which each UL40 variant might be ‘beneficial’ and how CD94/NKG2C adaptive NK cell number, phenotype and function are influenced by combinations of genetic variation in HLA-E, NKG2C and HCMV UL40 remain to be defined. They do remain potentially important for renal transplant and HSCT patients where control of HCMV is a clinical challenge.

Despite this, it is important to note that complete deletion of the *NKG2C* gene (*KLRC2*) occurs with approximately 4–8% homozygosity in Dutch, Japanese and Spanish cohorts. However, to date, there have been no reports correlating the *KLRC2 null* genotype with overt HCMV disease, and these donors can expand populations of ‘adaptive’ NK cells in the absence of NKG2C [118,119,120,121,122,123,124]. Expansions of CD94^+^NKG2C^+^ cells may, therefore, be a marker of host response to HCMV but are not necessarily critical for immune defence against the virus, and their relevance *in vivo* remains unclear.

### 6.3. NKG2D Ligands (NKG2DL)

NKG2D is an activating receptor found on virtually all NK cell subsets, as well as subsets of CD8^+^ αβ and γδ T-cells [125]. It is a C-type lectin receptor that signals through the adaptor protein DAP10. NKG2D ligands (NKG2DL) are upregulated on cells in response to various forms of cellular stress and are the HLA-I-related proteins, MICA, MICB and the UL16-binding proteins (ULBP) 1–6. These NKG2DL are upregulated on cells in response to various forms of cellular stress. In addition to infection with an intact virus, expression of just the immediate early (IE) proteins of HCMV drives cell surface expression of different NKG2DL [126,127].

The importance of NKG2D in the activation of cytotoxic lymphocytes is illustrated by the many HCMV gene products that target NKG2DL, currently standing at 9 different proteins or microRNAs (Figure 3). The first to be described was UL16, which retains MICB, ULBPs1 and 2 within the endoplasmic reticulum to prevent cell surface expression [128,129,130,131]. An HCMV-encoded microRNA, miR-UL112, binds to the MICB mRNA, thereby preventing its translation [132].

HCMV targets ligands for NKG2D (NKG2DL) at multiple levels. UL16 binds to MICB and ULBPs1, 2 and 5 and retains them within the ER. miR-UL112 prevents the translation of the MICB mRNA. UL142 binds to MICA and retains it within the trans-Golgi network (TGN). US18 and US20 target MICA for lysosomal degradation. US12 targets ULBP2 for lysosomal degradation. UL148A targets MICA for lysosomal degradation. US9 interferes with MICA*008 cell surface expression by targeting its GPI anchor. UL147A targets MICA*008 for proteasomal degradation.

Several other viral proteins can also target NKG2DL, including MICA. MICA is highly polymorphic, with 585 alleles currently described (compared to 307 alleles for MICB) that likely arise as a result of strong evolutionary selective pressure (https://hla.alleles.org/pages/genes/statistics/ (accessed on 31 March 2025)). MICA*008 is the most frequent MICA allele in most populations worldwide, except for Indigenous peoples of the Americas [133,134,135]. It has a frameshift that results in an altered transmembrane domain and a truncated cytoplasmic domain. It is thought the frameshift results in the protein acquiring a glycosylphosphatidylinositol (GPI) anchor, which confers distinct properties to the prototypic full-length MICA alleles [136].

HCMV targets both full-length and truncated MICA alleles. UL142 retains full-length MICA within the trans-Golgi network, as well as ULBP3 [137,138,139,140]. US18 and US20, members of the US12 family, cooperate in targeting full-length MICA for lysosomal degradation [126]. Another member of this family, US12, regulates ULBP2 cell surface expression, possibly by cooperating with UL16 [141]. UL148A targets full-length MICA alleles for lysosomal degradation [142], whilst US9 and UL147A target MICA*008. US9 targets MICA*008 via the cleft lip and palate transmembrane protein 1-like protein (CLPTM1L) and the GPI linkage, whereas UL147A targets it by maturational arrest and proteasomal degradation [143,144,145,146].

### 6.4. DNAM1/CD226 Ligands

DNAX accessory protein 1 (DNAM1) or CD226 is an NK activating receptor expressed by most NK cells, but also by T-cells and monocytes [147,148]. It recognises members of the Nectin/Nectin-like (Necl) family, including CD155 (Polio Virus Receptor/PVR/Necl-5) and CD112 (Nectin-2/PVRL2) [147]. HCMV encodes UL141 that retains CD155 within the ER and leads to the proteasomal degradation of CD112 by co-operating with US2, thereby preventing cell surface expression and activation of NK cells (Figure 4) [149,150,151]. DNAM1 is part of a ‘paired receptor’ family sharing ligands together with T-cell immunoreceptor with immunoglobulin and ITIM domain (TIGIT), poliovirus receptor-related immunoglobulin domain-containing protein (PVRIG) and CD96/T-cell-activated increased late-expression protein (TACTILE). Whereas DNAM1 has activating properties, TIGIT and PVRIG are inhibitory [147]. Despite this, in the context of HCMV infection, loss of DNAM1 signalling through downregulation of CD155 and CD112 is sufficient to provide an exceptionally strong inhibitory signal to a broad range of NK cells [149].

UL141 retains CD155 within the ER. UL141 (in magenta) co-operates with US2 to target CD112 for proteasomal degradation.

### 6.5. NKp30 and Its Ligands

NKp30 (also called Natural cytotoxicity triggering receptor 3/NCR3) is another of the NK activating receptors belonging to the natural cytotoxicity receptors along with NKp44 and NKp46 [152]. NKp30 contains an extracellular immunoglobulin domain and signals through the ITAM-containing adapter proteins CD3ζ and/or FcεRIγ. To date, only B7-H6 (NCR3LG1), a member of the B7 family of co-stimulatory molecules [41], and human leukocyte antigen-B-associated transcript 3 (BAT3/BAG6) have been described as ligands for NKp30. BAG6/BAT3 is released from cells as a secreted protein and triggers NKp30 [153], whereas galectin-3 (LGALS3) functions as a negative regulator of NKp30 signalling [154].

HCMV targets the NKp30 pathway via two main mechanisms (Figure 5). The first mechanism involves UL83 (pp65) binding directly to NKp30 to inhibit its function [155]. UL83 is a major component of the HCMV virion, as well as dense bodies which can be released from cells during infection. This may enable dense bodies and/or virions to impair NK function via NKp30 if particles become disrupted and release soluble pp65 [156]. The second mechanism involves US18 and US20, two members of the US12 family, which target B7-H6 for lysosomal degradation [141,157].

UL83 (pp65) binds directly to NKp30 and inhibits NKp30 signalling. US18 and US20 target the NKp30 ligand, B7-H6, for lysosomal degradation.

### 6.6. HCMV Targeting of the Extrinsic Apoptotic Pathway

In addition to the release of cytotoxic molecules by NK degranulation, NK cells can kill target cells through pro-apoptotic members of the tumour necrosis factor (TNF) superfamily, such as TNF-related apoptosis-inducing ligand (TRAIL) and Fas ligand/CD95L. TRAIL can either be membrane-bound on the NK cell or released as a soluble molecule. TRAIL induces apoptosis via its interaction with the death receptors TRAIL-R1 and TRAIL-R2 [54] but has also been implicated in the induction of NK cell degranulation [158]. CD95L triggers apoptosis by binding to CD95 (Fas). Both events result in a signalling cascade involving the activation of caspase-8 and the extrinsic apoptotic pathway.

This pathway is targeted by several HCMV-encoded immune evasins, notably UL141, UL36 and UL4 (Figure 6) [159,160,161]. In addition to inhibiting DNAM-1-driven NK cell activation (see Section 6.4), UL141 binds directly to TRAIL receptors and sequesters them within the ER, thereby preventing their trafficking to the cell surface and inhibiting TRAIL-mediated apoptosis of the HCMV-infected cell [159,162].

UL141 binds to and retains TRAIL-Rs within the ER. UL4 binds to TRAIL and prevents its binding to TRAIL-Rs and also inhibits NK activation. HCMV also downregulates cell surface expression of CD95/Fas by an unknown mechanism. UL36 and other HCMV-encoded proteins inhibit downstream caspase activation and apoptosis.

More recently, UL4, a secreted member of the RL11 gene family, was found to inhibit NK function by binding both soluble and membrane-bound TRAIL with high affinity [161]. It has been shown to act as a decoy receptor, preventing TRAIL-R engagement and blocking TRAIL-induced apoptosis and NK activation [161]. Importantly, UL4 may influence more than the HCMV-specific immune response [161]. As a soluble immune evasin, it also has the potential to dampen NK immunity systemically, contributing to the opportunistic infections observed clinically with HCMV infection [161].

In parallel, HCMV also targets the CD95 (Fas) signalling pathway, another member of the TNF receptor superfamily (TNFRSF). The cell surface expression of CD95 is downregulated in HCMV-infected cells, thus inhibiting CD95-mediated apoptosis. Currently, the HCMV gene responsible for this effect is unknown [163].

In addition, multiple HCMV genes, such as UL36, UL37, UL122/IE2, and UL45, block apoptosis by inhibiting downstream caspase activation [160,164,165,166]. These mechanisms act at or after death-inducing signalling complex (DISC) formation and likely affect both CD95 and TRAIL-R apoptotic pathways, contributing to the HCMVs complex strategy to thwart NK function and promote viral persistence.

### 6.7. Killer Immunoglobulin-Like Receptors (KIRs)

Killer immunoglobulin-like receptors (KIRs) are members of the immunoglobulin-like receptor superfamily [39]. The human KIR gene family consists of 13 genes (KIR2DL1, KIR2DL2/L3, KIR2DL4, KIR2DL5A, KIR2DL5B, KIR2DS1, KIR2DS2, KIR2DS3, KIR2DS4, KIR2DS5, KIR3DL1/S1, KIR3DL2, KIR3DL3) and 2 pseudogenes (KIR2DP1 and KIR3DP1) on chromosome 19 [39]. Many of the 13 KIR genes and their alleles currently do not have known ligands. KIRs can be either activating or inhibitory in nature, and the former is characterised by short cytoplasmic domains, represented with an ‘S’ in their name, e.g., KIR2DS1. Inhibitory KIRs, in general, possess a long cytoplasmic domain and contain an “L” in their name, e.g., KIR3DL1. Inhibitory KIRs are characterised by the presence of immunoreceptor tyrosine-based inhibitory motif (ITIM) in the intracellular cytoplasmic domain, whereas activating KIRs lack ITIMs, and instead bind the DAP12 adaptor protein, which contains an ITAM [39]. Individuals inherit a combination of KIR genes from their parents, resulting in highly diverse KIR genotypes, which are rarely (<2%) identical between unrelated individuals [167]. KIR genes are highly polymorphic, with ~600 alleles identified, second only to the human leukocyte antigen (HLA) genes in their variability [39]. KIR genotypes can contain a combination of genes associated with two recognised KIR haplotypes: A and B [168,169,170]. KIR haplotypes can be further divided into two regions, centromeric (Cen) and telomeric (Tel), based on the location of the locus of the KIR genes present in the haplotype [39]. Three centromeric regions have been identified (CenA, CenB1 and CenB2) and two telomeric regions (TelA and TelB) [39]. Haplotype A is composed of genes from the CenA and TelA regions, with KIR2DS4 representing the only activating KIR. In comparison, haplotype B is composed of genes from either the CenA/TelB, CenB/TelA or CenB/TelB regions, representing larger genetic diversity than haplotype A [171]. These haplotypes have both activating and inhibitory KIR genes [168,171]. However, haplotype B has more activating KIRs than haplotype A (KIR2DS1, KIR2DS2, KIR2DS3, KIR2DS5 and KIR3DS1 vs KIR2DS4) [39].

Highly polymorphic KIRs primarily bind equally polymorphic HLA-I [172]. Various specific KIR-HLA-I binding combinations have been identified as conferring both risk and protection in the context of viral infections, such as human immunodeficiency virus (HIV), hepatitis C virus (HCV), and HCMV. For example, the activating KIR KIR3DS1, in combination with HLA-B alleles that encode molecules with isoleucine at position 80 (HLA-B Bw4-80Ile), is associated with delayed progression to AIDS following HIV infection [173]. Immunocompromised patients homozygous for haplotype A or the HLA-B Bw4-80Ile allele are at a significantly greater risk of symptomatic HCMV disease following primary infection [169]. The expression of KIR3DL1 during reactivation of HCMV in renal transplant patients resulted in increased lysis of HCMV-infected fibroblasts [174]. In contrast, individuals with haplotype B motifs are associated with HCMV protection in renal transplants, liver transplants and hematopoietic stem cell transplant recipients with lymphoid disease [170,175,176,177]. Specifically, KIR2DS1, KIR2DS3 and KIR3DS1 genes and/or the presence of more than one activating KIR have been shown to reduce HCMV reactivation by 65% in patients receiving stem cell transplant in separate studies [177,178]. Additionally, the lack of KIR2DS2 and the presence of KIR2DL3 were risk factors for post-transplant HCMV infection in kidney transplant recipients [179].

It has been suggested that pressure on the immune system exerted by pathogens, such as HCMV, may have resulted in increased polymorphism of KIRs due to the parallel evolution of KIRs and herpesviruses in humans. HCMV causes stable expansions of specific NK subsets, and Beziat and colleagues demonstrated that in these expanded subsets, there was at least one inhibitory KIR capable of binding HLA-C [114]. Despite the importance of KIRs on NK function, the identification of specific KIR ligands encoded by HCMV has not yet been reported in the literature. However, there have been reports of expansions of NK subsets expressing specific KIRs resulting from HCMV infection, mainly in the transplant setting. Promisingly, van der Ploeg and colleagues demonstrated an HLA-dependent recognition by KIR2DL1 of HCMV-infected fibroblasts and responsiveness of KIR2DS1 to cells infected with variants of the HCMV TB40/E strain (Figure 7) [180]. Further work is needed to fully comprehend the role of KIRs in either the control of HCMV or its immune evasion involving specific KIR^+^ NK subsets.

HCMV can trigger the inhibitory KIR2DL1 in an apparently HLA-C-dependent manner. This effect appears to be mediated by an HCMV-encoded peptide. Specific variants of HCMV TB40/E were also able to trigger KIR2DS1, potentially through upregulation of HLA-C [180].

### 6.8. Disruption of the NK Immune Synapse

To be activated and kill their target cells, an “immunological synapse” (IS) must be formed between the NK cell and its target. The IS is highly structured and co-ordinates the ability of different signalling molecules to appropriately engage their receptors while permitting the directional secretion of lytic granules into the target cell [181]. Actin polymerization within the NK cell is critical to this process [181]. However, analysis of the HCMV-encoded UL135 protein revealed that actin polymerization in the target cell is also critical, and this requirement is exploited by HCMV (Figure 8). UL135 interacts with the actin-regulating WASP family Verprolin-homologous protein (WAVE2) complex via the Abelson interactor proteins-1 and -2 (ABI1 and ABI2) [182]. UL135-mediated relocalisation of WAVE2 results in the characteristic cytopathic effect of HCMV through loss of actin fibres. However, this also prevents IS formation because, under normal conditions, actin fibres in the NK cell intercalate with actin fibres in the target in order to form the IS. As a result, UL135 effectively inhibits the ability of all NK cells to successfully respond to infected cells. Adhesion molecules are also critical for IS formation. HCMV downregulates surface expression of CD58, a ligand for CD2, via UL148 as well as multiple integrins (ITGA1, ITGA2, ITGA4, ITGA7, ITGB1), via US2 (Figure 8) [151,183].

UL135 prevents actin polymerization, which impacts the formation of the immunological synapse. UL148 retains CD58 within the ER and prevents it from binding to its receptor, CD2, and stabilising the immunological synapse. US2 promotes the degradation of several integrins important for cell adhesion.

### 6.9. Evasion of ADCC

In addition to immune evasins targeting innate NK functions, HCMV also encodes multiple mechanisms that antagonise ADCC (Figure 9). At least four viral proteins are Fc receptors for IgG (RL11, RL12, RL13, UL119-UL118), some of which can antagonise ADCC through selectively binding and internalising antibodies that are simultaneously bound to cell surface antigens. This prevents their binding to CD16 and subsequent activation of NK cells (reviewed elsewhere; [184,185,186]). In addition to its retention of activating receptor ligands within the ER, the viral protein UL141 also binds and restricts cell surface transport of four of the five known HCMV glycoprotein complexes (trimer, pentamer, gB, and gpUL116). Mechanistically, this occurs via direct interactions with glycoproteins H and B (gH, gB), components common to these complexes and results in very low levels of cell surface-exposed antigen that reduce ADCC efficacy [187].

ADCC occurs when non-neutralising antibodies bind to (in this case) viral glycoproteins (in blue) expressed at the cell surface and trigger the CD16 receptor complex present on adaptive NK cells via their Fc region. HCMV encodes several viral Fc receptors (RL11, RL12, RL13, UL119-UL118) (in green), which can prevent this activation by binding the Fc portion of antibodies and preventing its availability to trigger ADCC. UL141 (in magenta) limits the amount of different HCMV glycoproteins on the cell surface, thereby limiting ADCC directed against these glycoproteins.

## 7. Overcoming HCMV-Mediated Immune Evasion by ADCC

Although HCMV is regarded as one of the most immune-evasive pathogens that infect humans, it is nevertheless well controlled by healthy immunity, at least until old age [188]. Clinical data shows that increased disease is observed in NK-deficient patients and in the transplant setting, where pre-existing immunity reduces the risk of HCMV reactivation (see Section 1). Therefore, although immune evasins enable virus persistence, they are not an insurmountable barrier to the generation of protective immunity, even if sterilising immunity is unlikely. Since NK cells are critical to virus control *in vivo*, and the expansion of adaptive NK cells correlates with protection (see Section 4), there is increasing interest in exploiting ADCC in vaccines and immunotherapies. A key prerequisite for this is the identification of viral cell surface antigens that effectively promote this response.

HCMV encodes several entry glycoproteins—the fusogen gB, a dimeric complex (gM/gN) and at least two receptor binding complexes, trimer (gH/gL/gO), and pentamer (gH/gL/UL128/UL130/UL131A), as well as a complex containing gpUL116 that is currently of unknown function. Neutralising antibodies targeting these glycoproteins have been extensively studied as therapeutics due to their established role in blocking viral entry into permissive host cells. These antibodies effectively neutralise cell-free virus infection, inhibit virus spread *in vitro* [189,190,191,192] and can be induced in infected individuals and in clinical trials by vaccination [193,194,195,196,197,198,199,200,201]. However, several lines of evidence indicate that neutralisation alone is insufficient to control HCMV *in vivo*. Despite raising high neutralising titres, no vaccine has yet met its primary clinical endpoint. Furthermore, in congenital infection, there is no correlation between pentamer/trimer antibody titres and transmission of HCMV from mothers to babies [202,203]. Clinical trials have also highlighted treatment with hyperimmune globulin (HIG), selected specifically for high neutralising titres or potently neutralising monoclonal antibodies, do not consistently induce strong antiviral effects or robust protection against HCMV [204,205,206,207,208,209].

This discrepancy between *in vitro* and *in vivo* results may arise due to the route of virus transmission in experimental as opposed to natural settings. *In vitro* experiments tend to use HCMV strains that contain either overt or subtle mutations in the genes RL13 and UL128/UL130/UL131A, which act to switch the virus to disseminate via the cell-free route as opposed to being highly cell-associated [210]. In contrast, clinical virus strains disseminate almost exclusively by direct cell-cell contact [211,212,213]. HCMV strains that retain wildtype RL13/pentamer sequences spread as cell-associated viruses and are highly resistant to neutralising antibodies [214]. Since HCMV disease occurs following the transplant of HCMV-infected organs or from an infected mother to the foetus, it is likely that cell-cell transmission underpins the initiation of infection, potentially explaining the failure of neutralising antibodies to prevent disease *in vivo*.

Studies of HCMV seronegative recipients vaccinated with entry glycoproteins demonstrate high neutralising titres but only weak ADCC responses—whereas responses from seropositive donors are far stronger [215]. Thus, there are clearly non-entry glycoproteins that dominate the ADCC response. An unbiased proteomics approach (quantitative temporal viromics; QTV [216]) identified 13 novel cell-surface proteins expressed by HCMV-infected cells during these early stages of infection, of which at least five could drive potent ADCC responses. Interestingly, these antigens were not structural glycoproteins but immune evasins (RL11, UL5, UL16, UL141 and US28) [216,217]. Monoclonal antibodies raised against one of these proteins were initially ineffective against HCMV-infected cells; however, engineering the Fc portion to increase their affinity to CD16 rendered them able to make NK cells responsive to HCMV-infected cells, thereby suppressing HCMV spread in viral dissemination assays (VDA) [217,218]. These antibodies were also unaffected by the presence or absence of the viral FcR decoys, suggesting that not only is HCMV susceptible to ADCC despite encoding immune evasins but that antibody engineering can be used to both enhance ADCC and circumvent viral countermeasures.

Interestingly, assessing ADCC responses in HCMV seropositive mothers and their babies showed that higher maternal sera ADCC activation is also associated with a lower risk of congenital CMV transmission [219]. Strikingly, ADCC responses correlated most strongly with anti-UL16 antibodies capable of binding CD16, supporting an important role for these novel ADCC targets *in vivo*. This is particularly fascinating given that antibodies with di-galactosylated Fc-glycans, which selectively bind FcRn and CD16, are superior at placental transfer [220,221]. This demonstrates that NK-activating antibodies may be preferentially selected and, therefore, more effective at protecting neonates from congenital CMV infection. Furthermore, neonates with cCMV expand a novel subset of CD8+ T cells, capable of ADCC, that are phenotypically very similar to adaptive NK cells [222]. Whilst distinct from adaptive NK cells, expansions of these T-cells in neonates were associated with superior antibody-dependent degranulation and cytokine production in a similar manner to NK-mediated ADCC. This suggests that antibody-based therapeutics that enhance ADCC may target non-NK cells as well to overcome immune evasion and control HCMV infection.

## 8. Future Perspectives

Several questions remain regarding the evasion of the NK cell response by HCMV and will need to be addressed in the future.

### 8.1. Why Does HCMV Encode So Many NK Immune Evasins?

HCMV is a persistent virus that establishes lifelong infection in the host. CD34^+^ haemopoietic stem cells are the site of HCMV latency with reactivation of productive infection, potentially expanding the pool of latently infected cells [223,224]. Therefore, HCMV has to persist in an environment surrounded by cytotoxic immune cells like NK cells, CD8^+^ αβ T-cells and γδ T-cells, all of which broadly express NK receptors. Indeed, HCMV-specific terminally differentiated effector memory T-cells (TEMRA)-like CD8^+^ αβ T-cells express many NK cell receptors and respond to a different set of signals to conventional CD8^+^ T-cells [225]. Although some NK receptors targeted by HCMV are expressed on nearly all NK cells (e.g., DNAM1, NKG2D), many others are expressed only on some subsets, stochastically, or only in response to certain cytokine stimulations. Mass cytometry analysis of 28 NK cell receptors identified 6000 to 30,000 phenotypic populations within an individual and more than 100,000 phenotypes across a range of donors [48]. Since it is the balance of signals from across all receptors that determines activation status, there is no single dominant pathway akin to the HLA-I-TCR interaction in CD8^+^ T-cells for control of NK activation. To evade the NK response, HCMV has evolved a broad panel of immune evasins to target different receptor pathways and thereby evade the broadest possible range of NK cell phenotypes under a wide range of different cytokine-stimulated conditions.

### 8.2. Have We Mapped All HCMV-Encoded Immune Evasins and the Full Range of Effects?

The currently identified HCMV-encoded immune evasins target two inhibitory receptor pathways (LIR-1/LILRB1 and CD94/NKG2A) and four pathways promoting NK activation (NKG2D, DNAM1/CD226, NKp30, TRAIL-R), plus an actin polymerization inhibitor and targeting of adhesion molecules (Table 1). These known evasins would appear to target quite a narrow subset of potential receptor pathways. There are multiple reasons why this could be the case (i) these pathways could be the dominant pathways required for NK activation, and HCMV does not require further immune evasins, or (ii) we have identified only the most “easily”-identifiable immune evasins. Situation (i) remains possible, and (ii) might reflect the approaches used for screening for NK inhibitors, e.g., readouts of NK activation like CD107a mobilisation and access to good quality reagents, e.g., antibodies directed against NKG2DLs. Future approaches might need to leverage cutting-edge omics techniques to look at the single-cell level and include other readouts of virus control (e.g., viral dissemination assays [226], cytotoxicity, immune synapse formation, and NK differentiation). Sequence variation within both host receptor-ligand pairing and virus genes also needs to be considered (see below), and evasins that work in cooperation targeting broad pathways to regulate NK function (e.g., the US12 family targeting multiple cell surface receptors or UL148 and UL148D down-regulating A disintegrin and metalloproteinase with thrombospondin motifs 17 (ADAM17), thereby stabilising NK inhibitors on the cell surface) [126,141,227]. In addition, the impact of HCMV-encoded immune evasins targeting NK cells on other immune cells which express these receptors has not been fully considered. Many receptors that are considered classical NK receptors, e.g., NKG2D, DNAM1 and KIRs, are also expressed on CD8^+^ and γδ T-cells and could influence their functional responses. Currently, the identified NK immune evasion molecules have generally been studied in the context of innate NK function, generally using bulk NK cell populations. Their influence on the function of adaptive NK cells, particularly in the context of ADCC, has not been examined. It is likely that to gain a full understanding of HCMV-encoded NK immune evasins investigation of the full range of NK-mediated functions will be required.

### 8.3. What Is the Role of Natural Sequence Variation in Both the Host and the Virus?

Host genes encoding for components of the immune system are under significant evolutionary selective pressure resulting from infectious microorganisms. The most variable genes in the entire human genome are those encoding HLA-I genes, with 1000 s of alleles identified (https://hla.alleles.org/pages/genes/statistics/ (accessed on 31 March 2025)). HLA-I will impact NK function through the triggering of various KIRs in an allele-specific manner (e.g., KIR3DL1 and HLA-A/B alleles containing the Bw4 epitope or KIR2DL1 and HLA-C2). There are also high levels of allelic variation in MICA/B, which may influence targeting by HCMV-encoded immune evasins—with specific immune evasins targeting either full-length or altered MICA*008 alleles. KIR often form paired receptors and are hyperallelic, therefore showing high levels of evolutionary selective pressure. To date, only an HLA-I-dependent triggering of KIR2DL1 by HCMV has been shown without the HCMV gene function responsible being mapped [180].

Although HCMV, as a virus with a DNA genome, does not display the same level of sequence variation as RNA viruses, there are genes that are designated as “hypervariable” (as reviewed in this Special Topic by Venturini and Breuer) [228,229,230]. These genes include RL5A, RL6, RL12, RL13, UL1, UL9, UL11, UL73, UL74, UL120, UL139 and UL146. Seven of these genes are members of the RL11 gene family RL5A, RL6, RL12, RL13, UL1, UL9 and UL11. Whilst they are commonly described as being “hypervariable”, this variation is not random. Instead, they fall into 6 to 14 genotypes, and variation within a genotype tends to be limited ([229,230], unpublished). HCMV seems to have speciated into viruses with distinct variants at these “hypervariable” loci. Another form of variation is a significant number of HCMV genes that are present with a disrupted ORF in a significant minority (>10%) of clinical isolates (RL12, RL13, UL1, UL9, UL11, UL40, UL111A, UL128, UL133, UL136, UL142, UL145, UL148, UL150A, IRS1, US6, US7, US9, US12, US13, US27) [228,229]. This implies that these HCMV genes are under ongoing evolutionary pressure and are negatively selected against in certain hosts, leading to disruption dependent on specific immunogenetics. Previous work has predominantly identified NK evasion functions using a limited range of HCMV strains. It is possible that functions could be missed due to sequence variation present between different viral strains, and therefore use of a broader set of representative viruses moving forward will be important.

### 8.4. Does Targeting NK Activation Using Non-Neutralising Antibodies Represent a Therapeutic Approach to Overcome HCMV-Mediation NK Cell Evasion?

Studies investigating the risk of congenital HCMV transmission have identified a correlation of non-neutralising antibody levels capable of ADCC to be predictive of a reduced risk of transmission [219,231]. A similar finding of the importance of antibodies triggering ADCC for protection from HCMV viraemia has been made in kidney transplant patients following vaccination with a recombinant glycoprotein B vaccine [232]. HCMV infection primes NK cells for the ability to carry out ADCC [222,233], suggesting that ADCC may represent a major immune mechanism in the control of HCMV infection in healthy individuals. Indeed, ADCC-triggering antibodies have been isolated from healthy individuals, which recognise the expression of cell surface expressed immune evasins (i.e., UL16, UL141 and US28) and are able to control HCMV infection *in vitro* [217], making them an attractive strategy for vaccination and immunotherapies.

## 9. Concluding Remarks

HCMV infection and NK cell phenotype and function appear to be intrinsically linked. The virus has co-evolved with the human immune system, developing a myriad of different immune evasins specifically to modulate NK activation. It is not clear whether there remain more NK evasins to be discovered and how individual evasins rank in importance versus one another. However, it is apparent that they have the potential to be significant therapeutic targets, which could be inhibited to boost the host’s immune system. The variation in virus/host immune evasion interactions could underpin the risk of disease, making them key biomarkers for severe pathogenesis. Non-neutralising antibodies, capable of stimulating NK cells via ADCC, appear to be another effective way of overcoming HCMV’s NK evasion properties and potentially treating HCMV disease.

## Figures and Tables

**Figure 1 pathogens-14-00629-f001:**
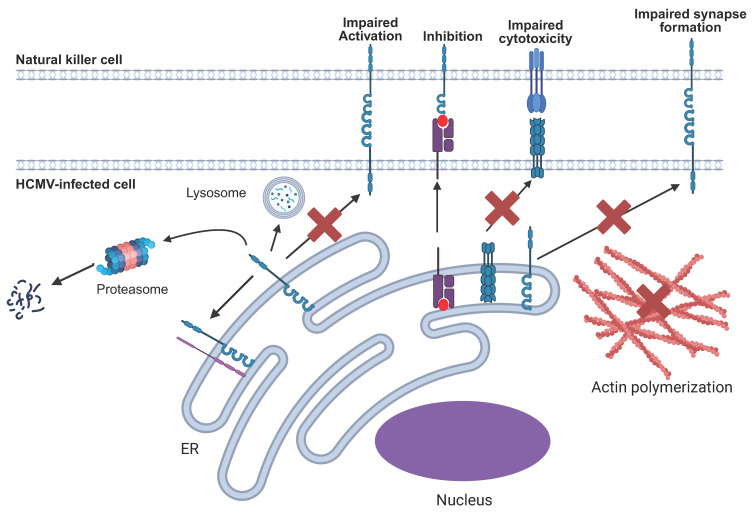
Summary of the different NK immune evasion mechanisms employed by HCMV.

**Figure 2 pathogens-14-00629-f002:**
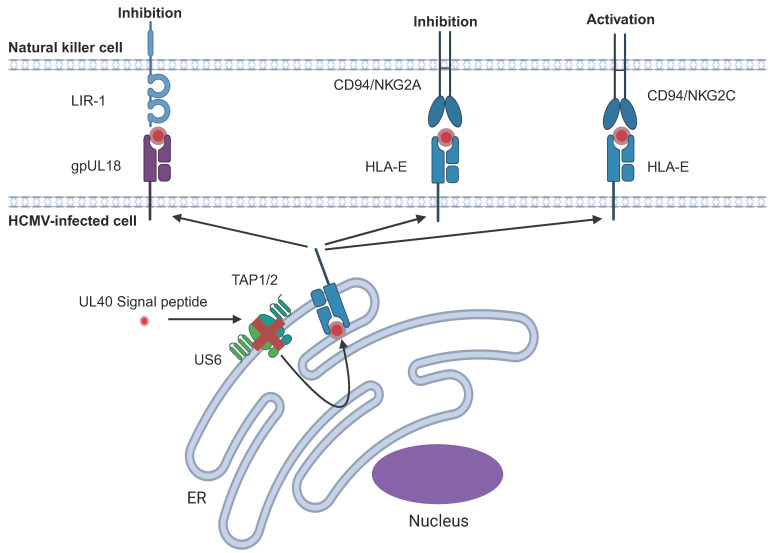
Upregulation of endogenous or HCMV-encoded inhibitory ligands.

**Figure 3 pathogens-14-00629-f003:**
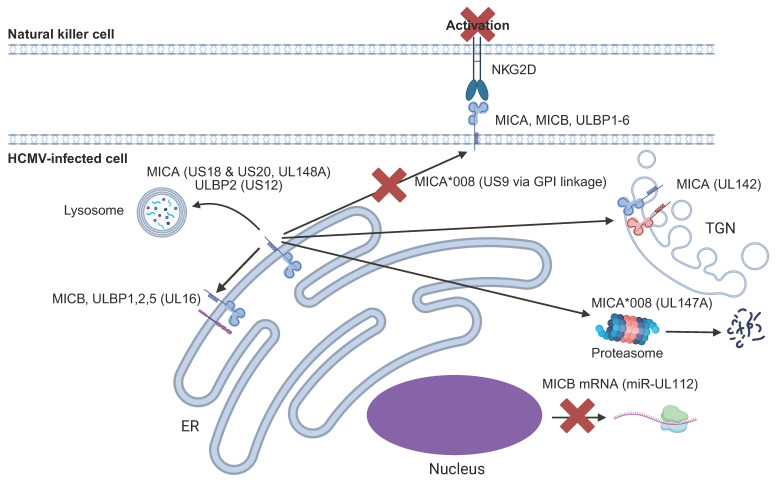
Regulation of NKG2D ligands by HCMV. * genetic alleles of different genes.

**Figure 4 pathogens-14-00629-f004:**
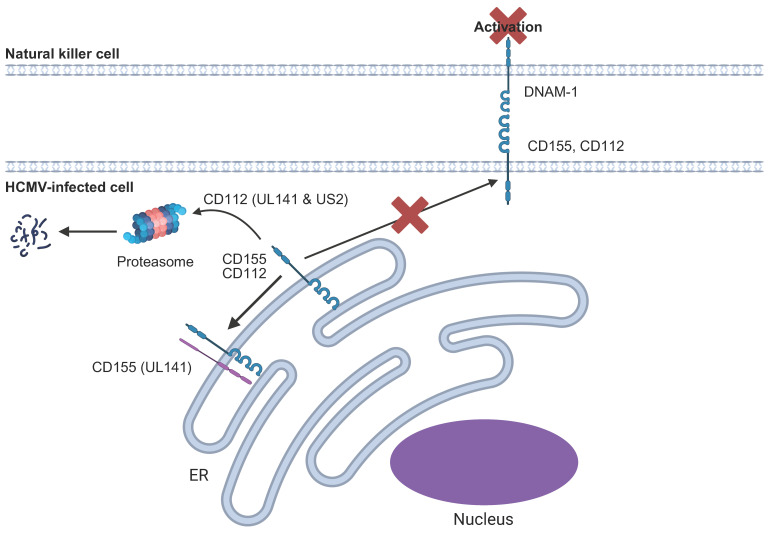
Targeting of DNAM1 ligands by HCMV.

**Figure 5 pathogens-14-00629-f005:**
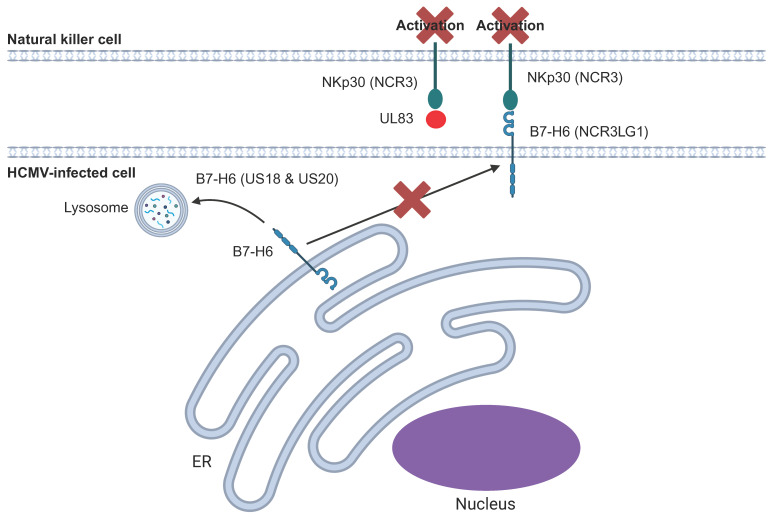
Regulation of NKp30 activation by HCMV.

**Figure 6 pathogens-14-00629-f006:**
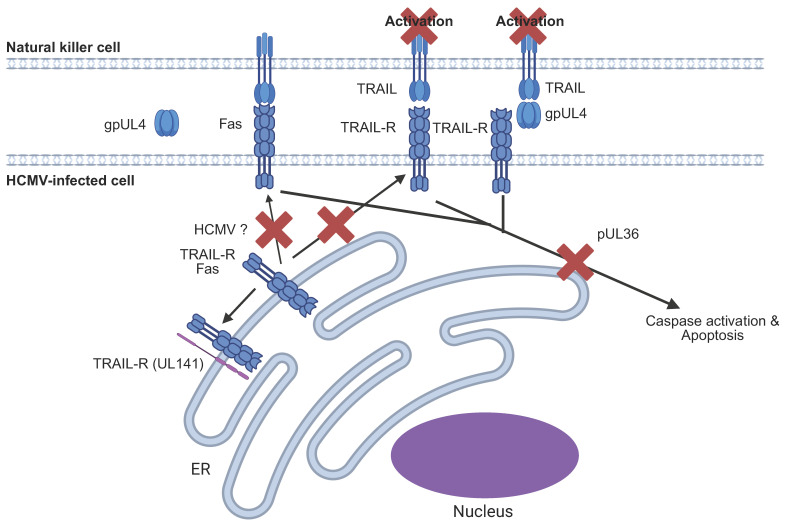
Inhibition of extrinsic cell death pathways by HCMV.

**Figure 7 pathogens-14-00629-f007:**
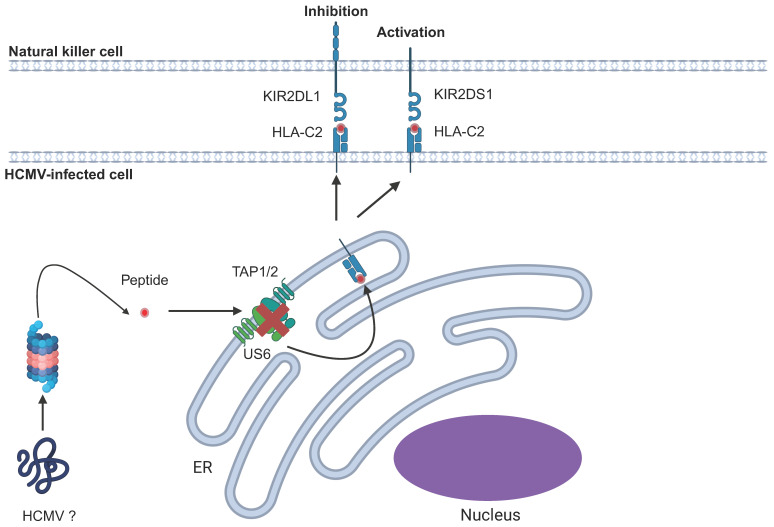
Effect of HCMV on KIRs.

**Figure 8 pathogens-14-00629-f008:**
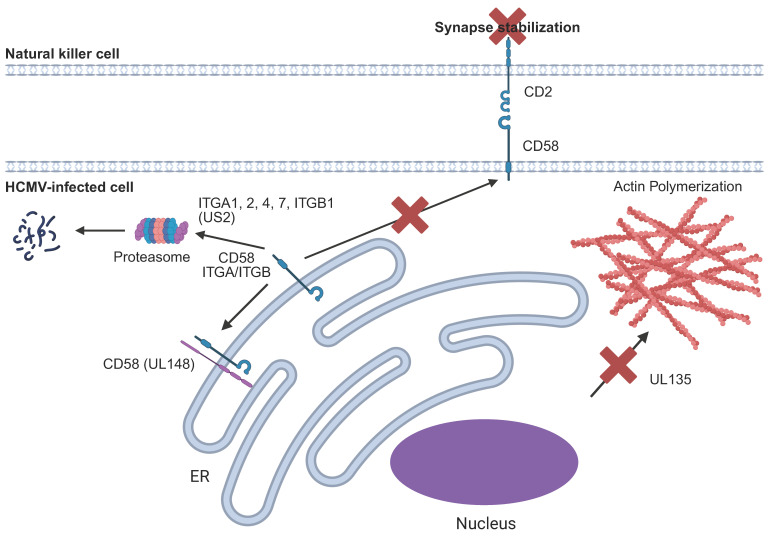
Inhibition of immunological synapse formation by HCMV.

**Figure 9 pathogens-14-00629-f009:**
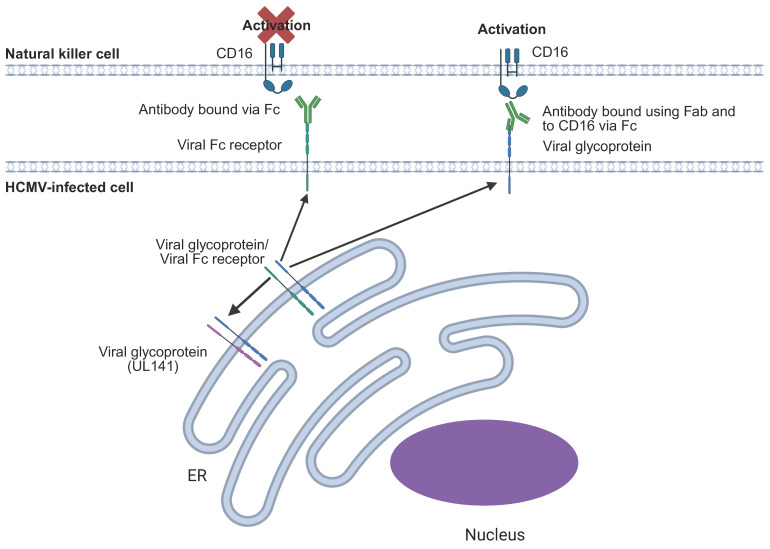
Antibody-dependent cellular cytotoxicity (ADCC) and its inhibition by HCMV.

**Table 1 pathogens-14-00629-t001:** Summary of HCMV-encoded NK immune evasins.

HCMV-Encoded Protein/microRNA	Cellular Target	Cellular Receptor	Mechanism	References
gpUL18	LIR-1/LILRB1	LIR-1/LILRB1	Direct binding to transmit an inhibitory signal	[98]
spUL40	HLA-E	CD94/NKG2A (CD94/NKG2C)	Up-regulation of surface HLA-E to transmit an inhibitory signal (NKG2C is the paired activating receptor)	[111,112,113]
UL16	MICB, ULBP1, ULBP2, ULBP5	NKG2D	Retains specific NKG2D ligands in the ER	[128,129,130,131]
UL142	MICA, ULBP3	NKG2D	Retains specific NKG2D ligands in the trans-Golgi	[137,138,139,140]
miR-UL112	MICB	NKG2D	Prevents MICB mRNA translation	[132]
US18 and US20	MICA	NKG2D	Targets MICA for lysosomal degradation	[126]
US12	ULBP2	NKG2D	Targets ULBP2 for lysosomal degradation, involvement of UL16?	[141]
UL148A	MICA	NKG2D	Targets MICA for lysosomal degradation	[142]
US9	MICA*008	NKG2D	Targets MICA*008 through its GPI anchor	[143]
UL147A	MICA*008	NKG2D	Inhibits MICA*008 maturation and proteasomal degradation	[146]
UL141	CD155	DNAM1	Retains CD155 within the ER	[149]
UL141 and US2	CD112	DNAM1	Targets CD112 for proteasomal degradation	[150,151]
US18 and US20	B7-H6	NKp30	Target B7-H6 for proteasomal degradation	[141,157]
UL83	NKp30	NKp30	Direct binding to inhibit the activating signal	[155]
UL141	TRAIL-R	TRAIL-R	Retains TRAIL-R within the ER to prevent the induction of apoptosis	[159]
UL4	TRAIL	TRAIL-R	Binds to TRAIL with high affinity to prevent pro-apoptotic signalling and NK activation	[161]
Unknown	CD95	CD95	Removes CD95 from the cell surface	[163]
UL36			Inhibitor of apoptosis	[160]
UL135	Actin polymerization	N/A	Prevents actin polymerization in the infected cell, thereby inhibiting the immunological synapse	[182]
UL148	CD58	CD2	Retains CD58 within the ER, preventing immunological synapse formation	[183]
UL141 and US2	Integrins		Targets various integrins for proteasomal degradation, inhibiting cell adhesion	[151]
Unknown	HLA-C-peptide	KIR2DL1	Contains a peptide which, when loaded onto HLA-C, triggers KIR2DL1 signalling	[180]
RL11	Fc portion of antibody	Anti-HCMV antibodies/CD16	Binds to Fc portion of antibody and prevents Fc-mediated signalling	[186]
RL12	Fc portion of antibody	Anti-HCMV antibodies/CD16	Binds to Fc portion of antibody and prevents Fc-mediated signalling	[185]
RL13	Fc portion of antibody	Anti-HCMV antibodies/CD16	Binds to Fc portion of antibody and prevents Fc-mediated signalling	[185]
UL119-UL118	Fc portion of antibody	Anti-HCMV antibodies/CD16	Binds to Fc portion of antibody and prevents Fc-mediated signalling	[186]
UL141	Viral glycoproteins	Anti-glycoprotein antibodies/CD16	Restricts cell surface expression of glycoprotein complexes, limiting the effect of anti-glycoprotein antibodies	[187]

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
