# Peer review of "Human Cytomegalovirus Immune Evasion of Natural Killer Cells: A Virus for All Seasons?"

_pathogens, 2025, doi:10.3390/pathogens14070629_

Round 1

Reviewer 1 Report

Comments and Suggestions for Authors

This manuscript is a comprehensive, well organized, and clearly presented review of NK cell-mediated mechanisms for control of HCMV infections and viral countermeasures to escape or mitigate NK control. This is an extremely complex field and the authors do a very good job of distilling an enormous and diverse body of literature into key points and concepts illustrated by relatively simple figures. Publication of this review will provide a valuable resource that may be particularly impactful as investigators engaged in HCMV vaccine development begin to grapple with recent findings that suggest neutralizing antibodies may not be sufficient to achieve robust protection.

While the figures are to be commended for their simplicity, clarity of the figures and figure legends could be better. Detailed suggestions are documented below.

Specific Comments:

  1. lines 159-160. It is unclear what "conventional (cNKs)" are.
  2. Line 168. Please clarify what "CD16 adaptor molecules" are.
  3. Lines 259-260 and perhaps elsewhere. Be consistent in using either "HLA-1" or "HLA-I".
  4. lines 282-283. Incomplete sentence.
  5. lines 412-413. Incomplete sentence.
  6. lines 435-437. Should be two sentences.
  7. line 494. Not clear what is meant by "sub-isolates of HCMV" or why it is important here - do some sub-isolates behave differently than others? "Isolate" usually refers to a virus cultured from clinical material. A sub-isolate would then be a clonal variant from an isolate. But per Fig. 7 it appears that this refers to variants of strain TB40/E, which is not an isolate.
  8. Line 534. "Overcoming" does not really capture the content of this section, which is not really about viral evasion mechanisms.
  9. Lines 562-565. This remains a controversial topic and this text is a bit one-sided. HIG has some clinical benefit in solid organ but not stem cell transplant. Similarly, monoclonal cocktails have shown efficacy in kidney (cited) but not stem cell transplant. Several studies failed to show efficacy for HIG in congenital infection, but one trial in which HIG was given very soon after diagnosis, more frequently, and at higher dose showed significant protection (PMID: 29947159).
  10. line 592. I prefer to restrict the use of "dissemination" to the in vivo spread of virus from one site to another. I don't think there is an in vitro equivalent.
  11. lines 608-611. Please clarify - these are T cells that mediate ADCC?
  12. line 716 and elsewhere. I find the term "non-neutralizing" imprecise and in some situations deceptive. It is accurate when describing antiviral mechanisms that do not involve virus neutralization, such a ADCC or ADCP. But when describing antibodies it can be deceptive and inaccurate - in a polyclonal mixture some of the antibodies that mediate ADCC/ADCP may not be able to neutralize, while others may be able to both neutralize and mediate ADCC/ADCP. Similarly, some monoclonals that mediate ADCC/ADCP may also be able to neutralize, while other may not.
  13. Fig. 1. Is the orientation of protein complexes in the ER membrane reversed? It seems the ectodomains should be in the lumen. It would help to label the ER and nucleus, and explain what the red fibrous material on the right represents. Figure legend is a bit of a run-on sentence.
  14. Fig. 2. This figure is a bit confusing as it appears to show the UL40-derived peptide utilizing the TAP transporter to enter the ER lumen even though TAP is blocked by US6. Transport of this peptide to the ER lumen it not blocked by US6. But does it still use TAP, or does it have some TAP-independent mechanism of transport to the ER?
  15. Fig. 3
    1. On the left NKG2DL is illustrated as a cytosolic globular protein and an arrow then suggests that it interacts with (?) or becomes a complex in the ER membrane (?). Other proteins are illustrated in the same way in subsequent figures. It is unclear what this is attempting to illustrate.
    2. Are the complexes in the ER membrane in the correct orientation (should the ectodomains be in the lumen)?
    3. Trans-Golgi is mentioned in the legend but does not appear to be illustrated in the figure.
  16. Fig. 4. Legend should perhaps indicate that UL141 is in magenta.
  17. Fig. 5. Is the orientation of B7-H6 in the ER membrane reversed?
  18. Fig. 6. Is the orientation of TRAIL-R and UL141 in the ER membrane reversed?
  19. Fig. 7
    1. Similar issues as in Fig. 2 regarding US6 and transport of the peptides.
    2. If this figure illustrates data from a particular paper it should be referenced in the legend.
  20. Fig. 8. Is the orientation of CD8 in the ER membrane reversed?
  21. Fig. 9
    1. Is the orientation of FCR in the ER membrane reversed?
    2. A bit confusing as the same blue symbol is used for both FCRs and glycoprotein targets of ADCC-mediating antibodies, but presumably FCRs are not retained in the ER by UL141?
  22. There are a lot abbreviations. Please confirm that each is used twice or more after being defined (exclusive of the abstract). If not then don't define as an abbreviation.
Comments on the Quality of English Language

Overall the writing is clear and concise. However, careful copy editing is necessary as there are numerous minor grammatical errors (misplaced commas, incorrect hyphenation, missing words, incomplete and run-on sentences, etc.).

Author Response

Reviewer 1

We thank the reviewer for their thoughtful and constructive comments, which we have addressed as detailed below.

Comment 1: lines 159-160. It is unclear what "conventional (cNKs)" are.

Response 1: We agree this was unclear and have added a sentence to try and clarify the term ‘conventional NK cells’. “Stochastic expression of this wide array of receptors means that there may be up to 30,000 different NK phenotypes in any one donor (47).Despite this variation, the bulk NK population in peripheral blood are referred to as conventional NK cells (cNK), whereas specific NK phenotypes are referred to by other names e.g. adaptive or tissue resident NK cells (48).”

Comment 2: Line 168. Please clarify what "CD16 adaptor molecules" are.

Response 2: The nature of the CD16 adapter proteins was unclear, they are CD3z and FceRγ. To clarify the text we have changed it to “For example, whereas cNKs express two CD16 adapter molecules, namely CD3z, a component of the T-cell receptor complex, and FceRγ, adaptive NK cells only express CD3z.”

Comment 3: Lines 259-260 and perhaps elsewhere. Be consistent in using either "HLA-1" or "HLA-I".

Response 3: We have now been consistent in use ‘HLA-I’ throughout the manuscript.

Comment 4: lines 282-283. Incomplete sentence.

Response 4: This paragraph and its meaning were altered during the editing process. We have changed it to “The non-classical HLA molecule HLA-E is upregulated at the cell surface following binding to conserved peptides derived from the leader sequences of other HLA-I molecules. Once there, it interacts with paired CD94/NKG2A and/or CD94/NKG2C receptors, thereby providing surveillance for endogenous HLA-I levels. Due to virus-induced targeting of the transporter associated with antigen processing (TAP) and subsequent loss of the native HLA-derived peptide, HCMV encodes a protein with a peptide mimic in its signal peptide (spUL40) which upregulates HLA-E independently of TAP (Figure 2) [106,107].”

Comment 5: lines 412-413. Incomplete sentence.

Response 5: We were unsure which sentence was being referred to here. We changed this sentence to “TRAIL induces apoptosis via its interaction with the death receptors TRAIL-R1 and TRAIL-R2 [49], but has also been implicated in the induction of NK cell degranulation [153].”

Comment 6: lines 435-437. Should be two sentences.

Response 6: We have split this longer sentence into the following ‘The cell surface expression of CD95 is downregulated in HCMV-infected cells, thus inhibiting CD95-mediated apoptosis. Currently the HCMV gene responsible for this effect is unknown [158]. ‘

Comment 7: line 494. Not clear what is meant by "sub-isolates of HCMV" or why it is important here - do some sub-isolates behave differently than others? "Isolate" usually refers to a virus cultured from clinical material. A sub-isolate would then be a clonal variant from an isolate. But per Fig. 7 it appears that this refers to variants of strain TB40/E, which is not an isolate.

Response 7: We agree this was unclear and have changed this nomenclature to “Promisingly, van der Ploeg and colleagues, demonstrated an HLA-dependent recognition by KIR2DL1 of HCMV-infected fibroblasts and responsiveness of KIR2DS1 to cells infected with variants of the HCMV TB40/E strain (Figure 7).”

Comment 8: Line 534. "Overcoming" does not really capture the content of this section, which is not really about viral evasion mechanisms.

Response 8: We appreciate the reviewer’s point, however antibody-mediated NK cell activation in response to a HCMV-infected cell has to overcome HCMV-encoded immune evasion mechanisms to be functional.

Comment 9: Lines 562-565. This remains a controversial topic and this text is a bit one-sided. HIG has some clinical benefit in solid organ but not stem cell transplant. Similarly, monoclonal cocktails have shown efficacy in kidney (cited) but not stem cell transplant. Several studies failed to show efficacy for HIG in congenital infection, but one trial in which HIG was given very soon after diagnosis, more frequently, and at higher dose showed significant protection (PMID: 29947159).

Response 9: We agree this statement may have been too one-sided. We have changed the text to “Clinical trials have also highlighted that treatment with hyperimmune globulin (HIG) selected specifically for high neutralising titres, or potently neutralising monoclonal antibodies, vary in their ability to induce strong antiviral effects or robust protection against HCMV” and added the suggested citation.

Comment 10: line 592. I prefer to restrict the use of "dissemination" to the in vivo spread of virus from one site to another. I don't think there is an in vitro equivalent.

Response 10: We understand the reviewer’s concerns and the term reflects our use of an assay termed a ‘virus dissemination assay’. We have changed the text to reflect this with the following “Monoclonal antibodies raised against one of these proteins were initially ineffective against HCMV-infected cells, however engineering the Fc portion to increase their affinity to CD16 rendered them able to make NK cells responsive to HCMV-infected cells, thereby suppressing HCMV spread in viral dissemination assays (VDA).”

Comment 11: lines 608-611. Please clarify - these are T cells that mediate ADCC?

Response 11: The reviewer is correct, we have changed the text to reflect this to “Furthermore, neonates with cCMV expand a novel subset of CD8+ T cells, capable of ADCC, that are phenotypically very similar to adaptive NK cells.”

Comment 12: line 716 and elsewhere. I find the term "non-neutralizing" imprecise and in some situations deceptive. It is accurate when describing antiviral mechanisms that do not involve virus neutralization, such a ADCC or ADCP. But when describing antibodies it can be deceptive and inaccurate - in a polyclonal mixture some of the antibodies that mediate ADCC/ADCP may not be able to neutralize, while others may be able to both neutralize and mediate ADCC/ADCP. Similarly, some monoclonals that mediate ADCC/ADCP may also be able to neutralize, while other may not.

Response 12: We understand the reviewer’s concern and agree antibodies may have neutralising or non-neutralizing function or both. However, the non-neutralizing antibodies referred to are directed against viral proteins which are either not present in the virion or not involved in virus entry, so are not capable of neutralisation.

Comment 13: Fig. 1. Is the orientation of protein complexes in the ER membrane reversed? It seems the ectodomains should be in the lumen. It would help to label the ER and nucleus, and explain what the red fibrous material on the right represents. Figure legend is a bit of a run-on sentence.

Response 13: We apologize and have reversed the protein in the ER. We have labelled the ER and nucleus in all figures and labelled the red fibrous material as polymerized actin. We have changed the legend to “HCMV NK immune evasion can be mediated by either the upregulation of endogenous or virus-encoded ligands for inhibitory NK receptors, downregulation of endogenous ligands for activating receptors. The latter may take place through either intracellular retention or degradation via the proteasome or lysosome. HCMV also impairs NK cell killing mechanisms by inhibiting cell death pathways at the ligand-receptor level or at the level of downstream caspase activation. It also interferes with the formation of the immunological synapse by targeting of adhesion molecules or actin polymerization.”

Comment 14: Fig. 2. This figure is a bit confusing as it appears to show the UL40-derived peptide utilizing the TAP transporter to enter the ER lumen even though TAP is blocked by US6. Transport of this peptide to the ER lumen it not blocked by US6. But does it still use TAP, or does it have some TAP-independent mechanism of transport to the ER?

Response 14: The UL40-derived peptide is loaded in a TAP-independent manner.

Comment 15: Fig. 3. On the left NKG2DL is illustrated as a cytosolic globular protein and an arrow then suggests that it interacts with (?) or becomes a complex in the ER membrane (?). Other proteins are illustrated in the same way in subsequent figures. It is unclear what this is attempting to illustrate.

Response 15: We were trying to describe protein expression and loading to the ER before subsequent transport within the cell. We agree this was unclear so have removed the ‘globular protein’ from all figures to avoid confusion, excepting the figure regarding KIRs, where it remains necessary.

Comment 16: Fig. 3. Are the complexes in the ER membrane in the correct orientation (should the ectodomains be in the lumen)?

Response 16: We have reversed the protein in the ER.

Comment 17: Trans-Golgi is mentioned in the legend but does not appear to be illustrated in the figure.

Response 17: We have added the TGN into this figure in order to match the legend.

Comment 18: Fig. 4. Legend should perhaps indicate that UL141 is in magenta.

Response 18: We have indicated that UL141 is in magenta in the figure legend.

Comment 19: Fig. 5. Is the orientation of B7-H6 in the ER membrane reversed?

Response 19: We have reversed the protein in the ER.

Comment 20: Fig. 6. Is the orientation of TRAIL-R and UL141 in the ER membrane reversed?

Response 20: We have reversed the protein in the ER.

Comment 21: Fig. 7. Similar issues as in Fig. 2 regarding US6 and transport of the peptides.

Response 21: During a lytic HCMV infection, US6 is expressed and TAP will be blocked. Therefore, the effect is presumably TAP-independent. We have unpublished data using a TAP-deficient cell line and the same KIR2DL1 suggesting this is the case.

Comment 22: Fig. 7. If this figure illustrates data from a particular paper it should be referenced in the legend.

Response 22: We have referenced the paper in the legend.

Comment 23: Fig. 8. Is the orientation of CD8 in the ER membrane reversed?

Response 23: We have reversed the protein in the ER.

Comment 24: Fig. 9. Is the orientation of FCR in the ER membrane reversed?

Response 24: We have reversed the protein in the ER.

Comment 25: Fig. 9. A bit confusing as the same blue symbol is used for both FCRs and glycoprotein targets of ADCC-mediating antibodies, but presumably FCRs are not retained in the ER by UL141?

Response 25: We have changed the viral FcR to a green colour and indicated the different colouring in the figure legend.

Comment 26: There are a lot of abbreviations. Please confirm that each is used twice or more after being defined (exclusive of the abstract). If not then don't define as an abbreviation.

Response 26: We have removed any abbreviations that are used less than twice from the abbreviation list.

Comment 27: Overall the writing is clear and concise. However, careful copy editing is necessary as there are numerous minor grammatical errors (misplaced commas, incorrect hyphenation, missing words, incomplete and run-on sentences, etc.).

Response 27: We have re-proofed the manuscript to remove these errors.

Reviewer 2 Report

Comments and Suggestions for Authors

This is a comprehensive and well-written review that thoroughly covers the strategies employed by Human Cytomegalovirus to evade Natural Killer cell-mediated immunity. The manuscript provides a strong overview of both innate and adaptive NK cell responses, outlines the major immune evasion mechanisms encoded by HCMV, and discusses the therapeutic relevance of antibody-dependent cellular cytotoxicity. The authors effectively synthesize a broad range of current research and present it in a clear, structured, and logical manner. However, some points have to be addressed in order to improve the manuscript.

1-Coverage of NK Cells in Cancer and HCMV-Related Oncomodulation.

The manuscript does not sufficiently address the role of NK cells in cancer, particularly in the context of HCMV-induced cancers (oncomodulation or oncogenesis). Given the growing body of literature implicating HCMV in modulating tumor biology and immune surveillance, this omission limits the completeness of the review.

Recommendation: Include a substantial chapter on the following:

  • The emerging concept of HCMV “oncomodulation”, describing how viral proteins influence the tumor microenvironment.
  • The role of NK cells in tumor immunosurveillance, and how HCMV-mediated immune modulation may reduce their effectiveness.
  • Relevant evidence showing NK phenotypic alterations in HCMV-positive cancer patients and the presence of HCMV DNA/proteins in tumors (e.g., glioblastoma, breast, colorectal cancer,..).
  • Therapeutic strategies that target both HCMV persistence and tumor immune evasion.
  • Some references to cite:
  1. Herbein, Georges. “The Human Cytomegalovirus, from Oncomodulation to Oncogenesis.” Viruses 10,8 408. 3 Aug. 2018, doi:10.3390/v10080408)
  2. Herbein G. (2022). Tumors and Cytomegalovirus: An Intimate Interplay. Viruses14(4), 812. https://doi.org/10.3390/v14040812)
  3. M López-Botet, A De Maria, A Muntasell, M Della Chiesa, C Vilches: Adaptive NK cell response to human cytomegalovirus: facts and open issues. Semin Immunol, 65 (2023), Article 101706

2-Manuscript structure and flow: The review has rich content, but the flow from discussion to conclusion to future directions could be slightly improved.

Author Response

Reviewer 2

We thank the reviewer for their thoughtful and constructive comments, which we have addressed as detailed below.

Comment 1: Coverage of NK Cells in Cancer and HCMV-Related Oncomodulation.

The manuscript does not sufficiently address the role of NK cells in cancer, particularly in the context of HCMV-induced cancers (oncomodulation or oncogenesis). Given the growing body of literature implicating HCMV in modulating tumor biology and immune surveillance, this omission limits the completeness of the review.

Recommendation: Include a substantial chapter on the following:

  • The emerging concept of HCMV “oncomodulation”, describing how viral proteins influence the tumor microenvironment.
  • The role of NK cells in tumor immunosurveillance, and how HCMV-mediated immune modulation may reduce their effectiveness.
  • Relevant evidence showing NK phenotypic alterations in HCMV-positive cancer patients and the presence of HCMV DNA/proteins in tumors (e.g., glioblastoma, breast, colorectal cancer,..).
  • Therapeutic strategies that target both HCMV persistence and tumor immune evasion.
  • Some references to cite:
  1. Herbein, Georges. “The Human Cytomegalovirus, from Oncomodulation to Oncogenesis.” Viruses 10,8 408. 3 Aug. 2018, doi:10.3390/v10080408)
  2. Herbein G. (2022). Tumors and Cytomegalovirus: An Intimate Interplay. Viruses, 14(4), 812. https://doi.org/10.3390/v14040812)
  3. M López-Botet, A De Maria, A Muntasell, M Della Chiesa, C Vilches: Adaptive NK cell response to human cytomegalovirus: facts and open issues. Semin Immunol, 65 (2023), Article 101706

Response 1: We agree with the reviewer that we had omitted this important and emerging area of HCMV in cancer. We have addressed this by adding in a paragraph in the introductory part of the manuscript, including referencing the two Herbein reviews and one from Soderberg-Naucler. The area of adaptive NKs in the third reference has been covered and relates more to the expansion of these cells in HCMV infection and their general role in tumour immunotherapy.

“HCMV proteins and DNA have also been detected in various tumors (including glioma, medulloblastoma, neuroblastoma, colorectal cancer, prostate cancer and breast cancer) (19, 20). It is currently unclear whether HCMV has a causal role in oncogenesis or whether it infects existing tumors and influences disease progression through oncomodulation. Longitudinal studies in a large cohort are necessary to define any causal link, akin to those that have linked Epstein-Barr virus infection to the development of multiple sclerosis (21).”

We feel that a more substantial chapter on HCMV oncomodulation and the role of NK cells in tumour surveillance goes beyond the scope of this review. The review is currently on the upper limit of other reviews in the journal/issue and widening its scope would lose the focus of the review. While we agree it is an important area, there are many reviews currently on the role of NK cells in cancer immunotherapy. Any effect of HCMV on tumour immunosurveillance would be inferred as, to the best of our knowledge, there are no papers specifically investigating the role of HCMV-encoded immune evasins in anti-tumour immune responses.

Comment 2: Manuscript structure and flow: The review has rich content, but the flow from discussion to conclusion to future directions could be slightly improved.

Response 2: We have tried to improve the flow from the manuscript text into the future directions and then into the concluding remarks.

Round 2

Reviewer 2 Report

Comments and Suggestions for Authors

The author answered to all my concerns.